# Human leukocyte antigen variants associate with BNT162b2 mRNA vaccine response
Martina Esposito [1], Francesca Minnai[1,2], Massimiliano Copetti [3], Giuseppe Miscio[3], Rita Perna[3], Ada Piepoli[3], Gabriella De Vincentis[3], Mario Benvenuto [3], Paola D'Addetta[3], Susanna Croci[4,5], Margherita Baldassarri[4,5], Mirella Bruttini[4,5,6], Chiara Fallerini[4,5], Raffaella Brugnoni [7], Paola Cavalcante[7], Fulvio Baggi [7], Elena Maria Grazia Corsini [7], Emilio Ciusani[7], Francesca Andreetta [7], Tommaso A. Dragani[8], Maddalena Fratelli [9], Massimo Carella[3], Renato E. Mantegazza[7], Alessandra Renieri [4,5,6] & Francesca Colombo [1] ✉

## Abstract

**Background** Since the beginning of the anti-COVID-19 vaccination campaign, it has become evident that vaccinated subjects exhibit considerable inter-individual variability in the response to the vaccine that could be partly explained by host genetic factors. A recent study reported that the immune response elicited by the Oxford-AstraZeneca vaccine in individuals from the United Kingdom was influenced by a specific allele of the human leukocyte antigen gene *HLA-DQB1*.

**Methods** We carried out a genome-wide association study to investigate the genetic determinants of the antibody response to the Pfizer-BioNTech vaccine in an Italian cohort of 1351 subjects recruited in three centers. Linear regressions between normalized antibody levels and genotypes of more than 7 million variants was performed, using sex, age, centers, days between vaccination boost and serological test, and five principal components as covariates. We also analyzed the association between normalized antibody levels and 204 HLA alleles, with the same covariates as above.

**Results** Our study confirms the involvement of the HLA locus and shows significant associations with variants in *HLA-A*, *HLA-DQA1*, and *HLA-DQB1* genes. In particular, the HLA-A*03:01 allele is the most significantly associated with serum levels of anti-SARS-CoV-2 antibodies. Other alleles, from both major histocompatibility complex class I and II are significantly associated with antibody levels.

**Conclusions** These results support the hypothesis that HLA genes modulate the response to Pfizer-BioNTech vaccine and highlight the need for genetic studies in diverse populations and for functional studies aimed to elucidate the relationship between HLA-A*03:01 and CD8+ cell response upon Pfizer-BioNTech vaccination.

## Plain language summary

It is known that people respond differently to vaccines. It has been proposed that differences in their genes might play a role. We studied the individual genetic makeup of 1351 people from Italy to see if there was a link between their genes and how well they responded to the BNT162b2 mRNA COVID-19 vaccine. We discovered certain genetic differences linked to higher levels of protection in those who got the vaccine. Our findings suggest that individual's genetic characteristics play a role in vaccine response. A larger population involving diverse ethnic backgrounds will need to be studied to confirm the generalizability of these findings. Better understanding of this could facilitate improved vaccine designs against new SARS-CoV-2 variants.

The response to the vaccine against COVID-19 is highly variable among vaccinated individuals, as reflected by the levels of antibodies detected in their serum after vaccination[1]. This phenotypic variability may be partly influenced by host genetic factors, as shown for other types of vaccines[2].

A recent genome-wide association study (GWAS) by Mentzer AJ et al. reported that the immune response triggered by the ChAdOx1 nCoV-19 (AZD1222, Oxford-AstraZeneca) vaccine in UK individuals was associated with a specific allele of the HLA-DQB1 gene[3], which encodes a human leukocyte antigen (HLA) molecule.

The HLA locus is highly polymorphic and the HLA allele frequencies differ substantially across populations, suggesting that the genetics of the response to anti-COVID-19 vaccine may vary depending on the population background. Moreover, the investigation of individuals receiving a different type of vaccine may reveal novel genetic factors involved in the production of anti-SARS-CoV-2 antibodies in response to vaccination.

In this study, we perform a GWAS of anti-SARS-CoV-2 antibody levels in 1351 Italian subjects who received two doses of the BNT162b2 (Pfizer-BioNTech) vaccine. Our analysis identifies variants in the HLA locus, as well as specific HLA alleles, associated with antibody levels.

## Methods

### Study cohort and ethical statement

For this study, individuals who received the two doses of BNT162b2 COVID-19 vaccine (as per guidelines, the boost dose was administered 21 days after the first dose) were recruited in three Italian hospitals: Fondazione IRCCS Istituto Neurologico Carlo Besta in Milan ($n = 344$), Azienda Ospedaliero-Universitaria Senese in Siena ($n = 802$), and Fondazione IRCCS Casa Sollievo della Sofferenza, San Giovanni Rotondo (FG) ($n = 384$). The recruitment period spanned from December 27th 2020 and May 15th 2021. The study was performed in accordance with the Declaration of Helsinki. The research was approved by the ethics committees of recruiting hospitals, namely, the University Hospital (Azienda ospedaliero-universitaria Senese) ethical review board, Siena, Italy (Protocol n. 16917, amendment n. 11, dated March 4th, 2021), the Ethics Committee of IRCCS Istituto Tumori "Giovanni Paolo II", Bari at Fondazione Casa Sollievo della Sofferenza, San Giovanni Rotondo (FG), Italy (Protocol n. 65, amendment n. 11, dated May 13th, 2021), and the Ethics Committee Regione Lombardia, Sezione Fondazione IRCCS Istituto Neurologico "Carlo Besta", Milan, Italy (Protocol n. 85, dated June 9th 2021). All participants provided written informed consent to take part in the study and granted permission to use their biological samples and clinical data for genetic research purposes.

### Sample and clinical data collection

Study data from Azienda Ospedaliero-Universitaria Senese in Siena, and Fondazione IRCCS Casa Sollievo della Sofferenza, San Giovanni Rotondo were collected and managed using REDCap electronic data capture tools[4] hosted at Azienda Ospedaliero-Universitaria Senese. Data from subjects from Fondazione IRCCS Istituto Neurologico Carlo Besta were independently collected and stored in a dedicated database. Collected personal and clinical information included data about age, sex, previous symptomatic or molecularly detected SARS-CoV-2 infection (that was an exclusion criteria), date of vaccination (first and second dose), date of blood withdrawal. Peripheral blood samples were collected for automated genomic DNA extraction, while serum samples were obtained for measurement of antibody levels, independently in each of the three recruiting centers. The quantification of anti-SARS-CoV-2-spike antibodies (IgG) was performed, as single measurement, by Abbott (at Azienda Ospedaliero-Universitaria Senese and Fondazione IRCCS Istituto Neurologico Carlo Besta) and Siemens (at Fondazione IRCCS Casa Sollievo della Sofferenza, San Giovanni Rotondo) tests, and the measurement units were converted in binding antibody units (BAU)/ml, following manufacturer instructions. IgG measurement was done at a median time of 40 days after the administration of the second vaccine dose.

### Genome-wide genotyping, data quality control and HLA alleles imputation

Genomic DNA from 1509 samples was genotyped using the Axiom Human Genotyping SARS-CoV-2 Research Array (Thermo Fisher Scientific, CA, USA) at the Functional Genomics facility of the Instituto de Investigaciones Biomédicas August Pi i Sunyer (IDIBAPS, Barcelona, Spain). Genotype calling was performed using Axiom Analysis Suite software (Thermo Fisher Scientific) following the best practice workflow (with the modified average call rate threshold ≥97) and data of passed samples ($n = 1474$) were exported for subsequent steps. Quality control of genotype was carried out using

PLINK2 software[5], as well as principal component analysis (PCA). In detail, autosomal variants with a genotyping call rate < 95%, minor allele frequency (MAF) < 1%, and a Hardy-Weinberg equilibrium test $P$-value $< 1.0 \times 10^{-10}$, were filtered out. We removed samples with call rate < 98% ($n = 20$), with sex inconsistencies both for PLINK2 and Axiom Analysis Suite software ($n = 42$), with excess of heterozygosity ($n = 2$), and duplicates or related individuals up to the third degree of relatedness ($n = 28$). In addition, four samples of non-European origin were excluded after PCA (Supplementary Fig. 1), as well as individuals with no full phenotypic data available (i.e., IgG levels or any of the covariates; $n = 22$), individuals treated with immunosuppressive drugs and outliers with very low IgG values ($n = 5$). A flow diagram with selected subjects for genetic analyses is shown in Supplementary Fig. 2.

Genotype imputation to the whole genome sequence was performed using the Minimac4 on the Michigan Imputation Server, setting GRCh38/hg38 as array build, HRC r1.1 2016 (GRCh37/hg19) as reference panel (consisting of approximately 65,000 haplotypes from individuals predominantly of European ancestry), and phasing the data with Eagle v2.4[6–9]. The imputed genotypes were then filtered to exclude rare variants (MAF < 1%) and SNPs with a low-quality imputation ($R^2$ info score ≤ 0.7)[10].

HLA alleles were imputed using Minimac4 on the Michigan imputation server (https://imputationserver.sph.umich.edu/index.html#!pages/home), using the Four-digit Multi-ethnic HLA v2 reference panel (since no European-specific HLA panel was available) and phasing data with Eagle v2.4[11]. The imputed genotypes were then filtered to exclude variants with a low-quality imputation ($R^2$ info score ≤ 0.7).

### Statistical analyses

Comparisons of available demographic and clinical variables (age, sex, and time between second dose and serological measurement of IgG) among the three groups of vaccinated individuals from the three recruiting centers were done using Kruskal-Wallis and chi-squared tests, for quantitative and binary variables, respectively. Differences in IgG levels, measured $30 \pm 5$ days after the second dose of vaccine, among the three groups of individuals, recruited in the three centers, were evaluated with Kruskal-Wallis test, too. A threshold for significance for these tests was set at $P$-value < 0.05.

Normality of the distributions of quantitative variables was checked with Shapiro-Wilk test. A $P$-value < 0.05 meant a non-normal distribution. IgG values were normalized by doing an inverse-normal transformation in R environment, using the formula reported by[12]: `(qnorm((rank(x, na.last = "keep") - 0.5) / sum(!is.na(x))))`.

Linear regression between normalized IgG values and sex, age at vaccination, center (coded as dummy variable), and time between vaccination and serological test was carried out with `glm()` function in R. This was the model formula:

$$Normalized\ IgG \sim age + sex + time\ between\ second\ dose\ and\ serological\ test + center$$

Beta coefficients, with standard errors (SE) and 95% confidential intervals (CI) were reported. A threshold for significance was set at $P$-value < 0.05.

To investigate the association between imputed genetic variants and anti-spike IgG levels (normalized BAU/ml values), genome-wide linear regression was carried out with PLINK2. The analysis included 7,339,393 variants (each one coded as 0, 1, 2 according with the increasing number of minor alleles in the genotype, following the additive model). Sex, age at vaccination, center (coded as dummy variable), the first 5 principal components (PCs), and the time interval between the second vaccine dose and the serological test served as covariates. This was the model formula:

$$Normalized\ IgG \sim genotype + age + sex + time\ between\ second\ dose\ and\ serological\ test + center + PC1 + PC2 + PC3 + PC4 + PC5$$

Beta coefficients, with standard errors (SE) and 95% confidential intervals (CI) were reported. Genome-wide standard significance threshold was set at $P$-value < $5.0 \times 10^{-8}$. The library qqman in R was used to draw Manhattan and QQ plots. Zoom plot of the selected region of chromosome 6 was done with locus.zoom() function in R.

Other three models, corrected for the top-significant variants on the chromosome 6 were run:

*Normalized IgG ~ genotype + age + sex + time between second dose and serological test + center + PC1 + PC2 + PC3 + PC4 + PC5 + rs1632893genotype*

*Normalized IgG ~ genotype + age + sex + time between second dose and serological test + center + PC1 + PC2 + PC3 + PC4 + PC5 + rs28366135genotype*

and

*Normalized IgG ~ genotype + age + sex + time between second dose and serological test + center + PC1 + PC2 + PC3 + PC4 + PC5 + rs1632893genotype + rs28366135genotype*

Linear regressions between imputed four-digit HLA alleles ($n = 204$) and the normalized IgG values were carried out with glm() function in PLINK2 (following the pipeline described in[13], using sex, age at vaccination, centre (coded as dummy variable), and the time interval between the second vaccine dose and serological test, as covariates. Multiple test correction, using the Benjamini-Hochberg method[14], was applied to calculate the false discovery rate (FDR). A threshold for significance was set at FDR < 0.01. Minor allele frequencies and linkage disequilibrium (LD) between variants or HLA alleles were calculated with PLINK.

## Reporting summary
Further information on research design is available in the Nature Portfolio Reporting Summary linked to this article.

## Results
In this study, we included 1351 individuals with no previous SARS-CoV-2 infection, who received the two doses of BNT162b2 COVID-19 vaccine and were recruited in three Italian hospitals: Fondazione IRCCS Istituto Neurologico Carlo Besta in Milan ($n = 306$), Azienda Ospedaliero-Universitaria Senese in Siena ($n = 689$), and Fondazione IRCCS Casa Sollievo della Sofferenza, San Giovanni Rotondo (FG) ($n = 356$; Table 1). The recruitment period spanned from December 27th 2020 and May 15th 2021. Participants with European origin, as determined by principal component analysis (PCA, Supplementary Fig. 1) were included in the genetic analyses. The cohort consisted of Italian vaccinees who were primarily hospital workers, with a predominant female representation (66.5%), and a median age of 48 years (range: 19-84). The measurement of IgG levels was performed at a median time of 40 days after the administration of the second vaccine dose

(interquartile range, IQR = 68). IgG levels ranged from 12.64 to 6056 BAU/ml, with a median of 801.5 BAU/ml.

The three series were quite different in terms of sex distribution (chi squared test $P$-value = 0.0004): indeed, although the female sex was predominant in all the three series, Fondazione IRCCS Casa Sollievo della Sofferenza, San Giovanni Rotondo (hereafter SGR) recruited more male subjects (~42%) than the other recruiting centers (approximately 30%). The three series also differed for age distribution (Kruskal-Wallis test $P$-value < 0.0001) with SGR individuals being older than the others and Siena series having the lowest median age value. The most relevant difference between the series regarded the interval time (in days) between second vaccination dose and serological IgG measurement (Kruskal-Wallis test $P$-value < 0.0001). Indeed, IgG levels in Siena series were not measured 30 ± 5 days after the vaccine boost as, instead, it was done in the other two series.

Since IgG values were not normally distributed (Shapiro-Wilk test $P$-value < 0.0001), we applied the inverse normal transformation to these values (Supplementary Fig. 3). Then we performed a multivariable linear regression model between normalized IgG values, age, sex, recruiting centers, and time between vaccination and measurement of IgG levels (Table 2). Normalized IgG values inversely correlated with age at vaccination and time (in days) passed between vaccination and serum collection for antibody measurement. Antibody quantity, instead, was not significantly different between females and males. As expected, we observed lower levels of IgG in individuals from Siena than those recruited in the other two cohorts; the median IgG levels of Siena subjects differed of 902.5 and 1,078.5 BAU/ml from the median of Milan and SGR individuals, respectively. Nonetheless, it is important to note that, when we considered only those subjects whose antibodies were measured 30 ± 5 days after the second vaccine dose, we observed no significant differences in IgG levels among the three groups of individuals from the three centers (median values: 1160, 1395, and 1489 in Siena, Milan and SGR, respectively; Kruskal-Wallis test $P$-value > 0.05). This suggests that the differences between the three centers were more likely due to the discrepancies in the time intercourse between vaccination and IgG measurement, rather than to differences among operators and tests.

We carried out a genome-wide association analysis between normalized IgG levels and the imputed genotypes of 7,339,393 variants, including in the linear regression model sex, age at vaccination, recruiting center, the first 5 principal components (PCs), and the time interval between the second vaccine dose and the serological test as potential confounders. The results are reported in the Manhattan plot, shown in Fig. 1. A statistically significant signal was identified on chromosome 6, in the HLA locus, with 144 variants associated with a nominal $P$-value < $5.0 \times 10^{-8}$ (Supplementary Data 1). These variants spanned a region from 29.7 Mbp to 32.6 Mbp (according to human genomic build GRCh37/hg19) and the lead variant, rs1632893, mapped less than 4 kb ahead the *HLA-A* gene (beta = 0.28, SE = 0.044, 95% CI 0.20 – 0.37, $P$-value = $1.6 \times 10^{-10}$). A suggestive signal below the genome wide significance threshold was also observed on chromosome 2. The lead variant, rs11692649 at position 183,266,641, is an intronic variant of *PDE1A* gene (beta = −0.17, SE = 0.033, 95% CI −0.24 to −0.11, $P$-value = $1.87 \times 10^{-7}$).

**Table 1 | Personal and clinical characteristics of vaccinated subjects included in the genetic analyses and comparison between the three series of vaccinated subjects**

| Characteristic | Total ( = 1351) | Milan ( = 306) | Siena ( = 689) | SGR ( = 356) | *P*-value |
|---|---|---|---|---|---|
| Age (years), median (range) | 48 (19–84) | 47 (25–84) | 43 (19–78) | 54 (21–67) | <0.0001[a] |
| Sex, *n* (%) | | | | | |
| male | 452 (33.5) | 86 (28.1) | 218 (31.6) | 148 (41.6) | 0.0004[b] |
| female | 899 (66.5) | 220 (71.9) | 471 (68.4) | 208 (58.4) | |
| Days between vaccination and serological test, median (IQR) | 40 (68) | 30 (5) | 97 (37) | 30 (0) | <0.0001[a] |
| Serum Ab anti-SARS-CoV2 (BAU/ml), median (IQR) | 801.5 (1271.54) | 1343 (1635.93) | 440.5 (1330.01) | 1519 (569.5) | <0.0001[a] |

*SGR* San Giovanni Rotondo, *IQR* Interquartile range; BAU, binding antibody units.
[a]Kruskal-Wallis test.
[b]chi-squared test.

Zooming in the locus on chromosome 6, we observed that there was not a single association signal (Fig. 2). Indeed, there were other regions, in addition to the one led by rs1632893 at position 29,905,193, although this locus has the highest number of significantly associated variants ($n = 140$), spanning from 29.7 Mbp to 30.2 Mbp. For all these variants, that are in high ($r^2 > 0.7$) linkage disequilibrium among each other (except for eight of them), increasing number of minor alleles in genotypes was associated with highest levels of IgG. An additional peak of association (beta = 0.27, SE = 0.048, 95% CI: 0.17 – 0.36, $P$-value = $4.5 \times 10^{-8}$) was led by rs28366135 (at position 31,364,105), that maps less than 40 kbp upstream the *HLA-B* gene, together with rs2428479 at position 31,361,110 (beta=0.27, SE = 0.048, 95% CI: 0.17 – 0.36, $P$-value = $4.6 \times 10^{-8}$). Also for these variants, we observed that individuals homozygous for the minor alleles or heterozygous had higher levels of IgG than those homozygous for the major alleles. Then, we

observed other two variants (rs454875, beta = −0.23, SE = 0.042, 95% CI −0.32 to −0.15, $P$-value = $3.4 \times 10^{-8}$; rs9272454, beta = −0.17, SE = 0.031, 95% CI: −0.23 to −0.11, $P$-value = $4.2 \times 10^{-8}$) in positions 32,213,008 and 32,605,525, respectively. The latter was an intronic variant of *HLA-DQA1*. Another variant near to these two and just below the genome-wide significance threshold was rs28688207 (beta = −0.28, SE = 0.052, 95% CI: −0.38 to −0.18, $P$-value = $5.5 \times 10^{-8}$), a splice acceptor variant of *HLA-DQB1* gene. Differently from the previous variants, the minor alleles of these latter three was significantly associated with the lowest IgG levels.

We calculated the linkage disequilibrium (LD) between rs1632893 and the variants in the other two regions, and we observed that they were not in LD (D' = 0.14 and $r^2$ = 0.00033 with both rs28366135 and rs2428479; D' = 0.34 and $r^2$ = 0.0026 with rs454875, D' = 0.0020 and $r^2$ = $1.1 \times 10^{-6}$ with rs9272454, and D' = 0.086 and $r^2$ = $9.4 \times 10^{-5}$ with rs28688207; Supplementary Data 1), suggesting that the three signals are independent. Indeed, in a linear regression analysis testing the same model as above, but with the genotype of rs1632893 as an additional covariate, we observed that the other signals (rs28366135, rs2428479, as well as other variants in *HLA-B* locus, and rs9272454 in *HLA-DQA1*) remained statistically significant (Fig. 3A). The same analysis was done using rs28366135 as covariate, instead: in this case only the association signals in the *HLA-A* locus, led by rs1632893 remained significant (Fig. 3B). Finally, in a linear regression model where both rs1632893 and rs28366135 genotypes were added to the other covariates, the signal on chromosome 2 (rs11692649) became statistically significant (beta = −0.18, SE = 0.032, 95% CI −0.24 to −0.11, $P$-value = $4.9 \times 10^{-8}$), whereas the SNPs in *HLA-DQA1* and *HLA-DQB1* genes had $P$-values below the GWAS significance threshold (Supplementary Fig. 4).

Then, we imputed four-digits HLA alleles ($n = 204$) using the genotyping data of our 1351 individuals and we reported their frequencies in our series in Supplementary Data 2. We analyzed, in a linear regression model with the same covariates included in the GWAS, the association between the normalized IgG levels and the HLA alleles. We observed that

**Table 2 | Multivariable linear regression among normalized IgG values and personal and clinical information**

|  |  | beta | SE | 95% CI lower | 95% CI upper | *P*-value |
|---|---|---|---|---|---|---|
| age |  | −0.12 | 0.13 | −0.016 | −0.0087 | $5.0 \times 10^{-11}$ |
| center | Milan | 1 |  |  |  |  |
|  | SGR | 0.078 | 0.061 | −0.042 | 0.20 | 0.20 |
|  | Siena | −0.18 | 0.078 | −0.34 | −0.032 | 0.018 |
| sex | male | 1 |  |  |  |  |
|  | female | 0.066 | 0.045 | −0.023 | 0.15 | 0.15 |
| time between 2nd dose and IgG test |  | −0.015 | 0.00099 | −0.017 | −0.014 | $< 2.0 \times 10^{-16}$ |

*SE* standard error, *CI* confidential interval.

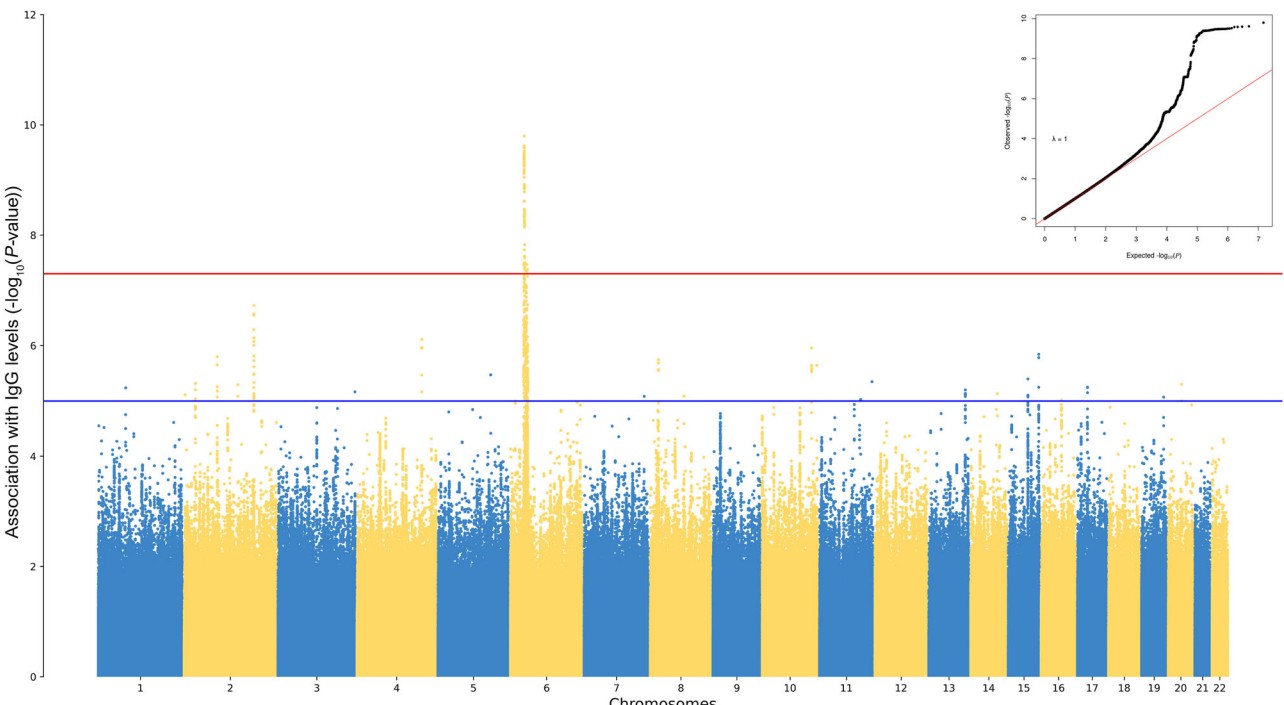

**Fig. 1 | A locus on chromosome 6 is strongly associated with anti-spike IgG levels.** Manhattan plot of the results of the GWAS between anti-spike inverse-normalized IgG values and 7,339,393 imputed variants, tested in a linear regression model, using sex, age at vaccination, center, the first 5 principal components (PCs), and the time interval between the second vaccine dose and the serological test as covariates. SNPs are plotted on the x-axis according to their genomic position (GChr37, hg19 release), and $P$-values ($-\log_{10}P$) for their association with IgG levels on the y-axis. The horizontal red line represents the threshold of genome-wide significance ($P$-value $< 5.0 \times 10^{-8}$). In the up-right corner is shown the Q-Q plot of observed and expected $P$ values. Genomic inflation factor ($\lambda$) is reported.

**Fig. 2 | Multiple signals of association in the HLA locus suggest a multi-gene control of IgG production after vaccination.** Zoom plot of the locus on chromosome 6 identified in the GWAS. Plots span the region from 29 Mbp to 33 Mbp, containing HLA genes and all the analyzed imputed variants. SNPs are plotted on the x-axis according to their position on chromosome 6, and $P$-values ($-\log_{10}P$) for their association with IgG levels are plotted on the y-axis. Horizontal red dashed line represents the threshold of significance ($P$-value $< 5.0 \times 10^{-8}$) whereas the blue one represents a suggestive threshold ($P$-value $< 1.0 \times 10^{-6}$). Dot color represents the level of linkage disequilibrium, expressed as $r^2$ between each SNP and the lead variant (rs1632893, purple diamond).

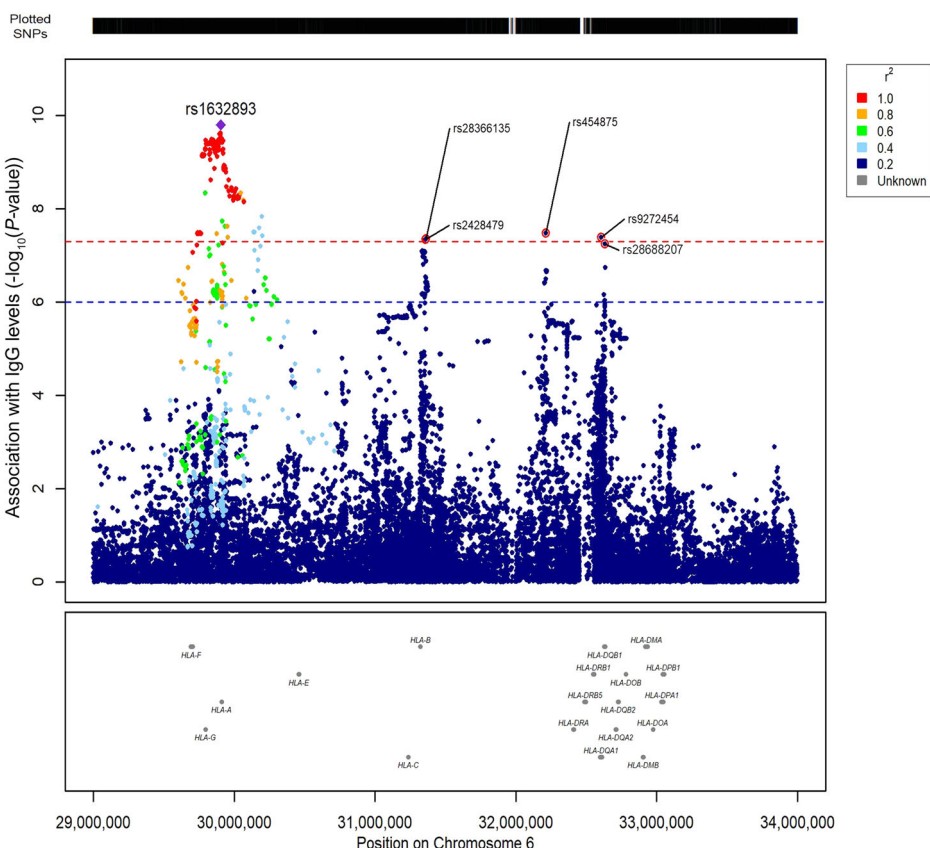

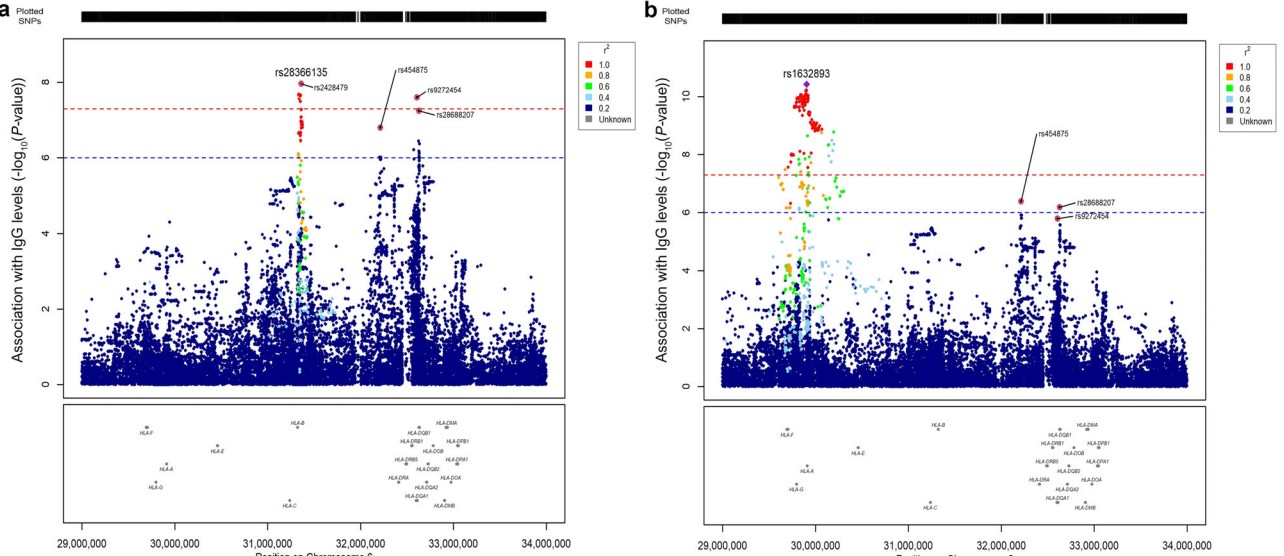

**Fig. 3 | Zoom plots of the locus on chromosome 6 after adjustment for rs1632893 or for rs28366135.** The two panels show the results of the analyses adjusted for rs1632893 (**a**) or for rs28366135 (**b**). Plots span the region from 29 Mbp to 33 Mbp, containing HLA genes and all the analyzed imputed variants. SNPs are plotted on the x-axis according to their position on chromosome 6, and $P$-values ($-\log_{10}P$) for their association with IgG levels are plotted on the y-axis. Horizontal red dashed line represents the threshold of significance ($P$-value $< 5.0 \times 10^{-8}$) whereas the blue one represents a suggestive threshold ($P$-value $< 1.0 \times 10^{-6}$). Dot color represents the level of linkage disequilibrium, expressed as $r^2$ between each SNP and the lead variant (purple diamonds).

12 HLA alleles were significantly associated with IgG levels at FDR < 0.01 (Supplementary Data 2). The two top-significant alleles (i.e., HLA-A*03:01 and HLA-C*12:02) belonged to class I major histocompatibility complex (MHC), together with other two alleles (HLA-B*52:01 and HLA-A*29:02). Instead, the other eight alleles significantly associated with IgG levels were MHC class II molecules (HLA-DQB1*06:01, HLA-DRB1*15:02, HLA-DQB1*02:01, HLA-DRB1*14:01, HLA-DQA1*01:01, HLA-DQA1*02:01, HLA-DQB1*05:03, and HLA-DRB1*07:01). The frequencies of these alleles are quite different, as expected (e.g., HLA-DQB1*02:01, HLA-DQA1*01:01, HLA-DQA1*02:01, HLA-A*03:01, and HLA-DRB1*07:01 were more frequent than HLA-C*12:02, HLA-DQB1*06:01, HLA-DRB1*15:02, HLA-DQB1*05:03, HLA-DRB1*14:01,

HLA-B*52:01 and HLA-A*29:02). Since some of these alleles were in LD, we reported the $R^2$ and D' values, together with their frequencies, in Supplementary Table 1.

## Discussion

In this GWAS, carried out in 1351 Italian individuals vaccinated with two doses of BNT162b2 anti-COVID-19 vaccine, we identified several variants on chromosome 6, in the HLA locus, associated with anti-SARS-CoV-2 IgG levels in serum, at a genome-wide statistically significant level. Our results independently validated the finding by Mentzer et al[3]. of an important role of HLA locus in the modulation of levels of anti-spike IgG after vaccination. Differently from that study, our results were obtained from Italian patients undergone to a different type of vaccine and, most interestingly, we found different HLA alleles, significantly associated with anti-spike IgG levels, from that identified in the UK population, suggesting the involvement of multiple genes, in this same HLA locus, in the modulation of immunogenic response against the anti-COVID-vaccine. Several studies have reported association of HLA variants and genotypes with different COVID-19 outcomes, even though with some discrepancies, and differences in the immune responses against SARS-CoV-2 infection (as reviewed in[15,16]). Several molecular mechanisms underlying HLA-mediated modulation of vaccine-induced immunity have been hypothesized and are currently under investigation. Here, we observed associations of eight MHC Class II molecules with COVID-19 vaccine immunogenicity. Of note, one of these HLA alleles was HLA-DQB1*06:01, in agreement with what already reported in[3]. Indeed, it has been shown that distinct Spike peptides binds preferably HLA-DQB1*06 alleles driving increased spike-specific memory B cell responses and higher antibody titers[3]. To our knowledge, here we first report that MHC Class I molecules associate with anti-spike IgG after vaccine immunization. It has been shown that anti-SARS-CoV-2 mRNA vaccine induced both neutralizing antibodies and CD8 + T cell responses[17]. Thus, we envisage a helper role of CD8+ follicular T cells in promoting class-switch antibody in B cells and amplifying antibody response, via both HLA binding and non-HLA related effects such as increased production of selected cytokines including IL-21[18–20]. Future studies are needed to investigate more in depth the relationship between HLA-A*03:01 allele, as well as other MHC class I molecules, and CD8 + T cell response upon anti-COVID-19 vaccination.

Interestingly, a previous GWAS reported a significant association between the HLA-A*03:01 and adverse events after vaccination with BNT162b2 Pfizer-BioNTech vaccine in a series of more than 3,500 Americans individuals[21]. They hypothesized that stronger adverse events, in vaccinees with HLA-A*03:01 allele, were due to a stronger activation of CD8 + T cells. Although the mechanisms involved in vaccine toxicity and efficacy might be quite different, that finding, together with ours, point to a pivotal role of HLA-A*03:01 in the response to Pfizer-BioNTech vaccine, possibly through the activation of CD8 + T cells.

Other HLA alleles, both of class I and class II MHC, for a total of 12 alleles, resulted significantly associated with IgG levels in our analysis. These alleles have different frequencies in our series ranging from 2% to 19%, as expected in Italians[22]. Also, some of them showed strong linkage disequilibrium (e.g., HLA-C*12:02 with HLA-B*52:01 and HLA-DRB1*15:02 with HLA-DQB1*06:01) and this complexity challenged the result interpretation.

Nonetheless, our findings shed light on the roles of both MHC class I and II molecules in the anti-COVID-19 vaccination response. They open further investigations aimed at deeply clarifying the contribution of the identified molecules to the individual protection toward COVID-19, and particularly the involvement of CD8 + T cells in B cell response and antibody production following mRNA vaccination. Protective effects of the HLA alleles we identified against SARS-CoV-2 infection after vaccination also deserve to be explored in further studies. Heterogeneity in the three subject series represented a limitation of the study, but it was overcome by considering all the possible confounder variables as covariates in the regression models.

Overall, our results provide further evidence, together with that provided by Mentzer et al.[3], for a genetic regulation of the response to anti-COVID-19 vaccines, mediated by the HLA locus. These results, in an independent European population, strengthen the importance to investigate these associations in individuals of different origin. The identification of specific HLA alleles conferring different ability to produce anti-spike IgG after vaccination with anti-COVID vaccine can be of clinical utility for tailoring vaccination campaign, especially in most fragile subjects. In addition to COVID-19, these results may stimulate geneticists to explore the genetics of the response to other type of vaccines, against different diseases, in view of a precision vaccination medicine supported by vaccinogenomics.

## Data availability

Raw genotyping data are not openly available to preserve individuals' privacy under the European General Data Protection Regulation. They are available from the corresponding author upon reasonable request. Data are located in controlled access data storage at Institute for Biomedical Technologies of the National Research Council. GWAS summary statistics are available in the GWAS catalog (accession number: GCST90305767). These data were used to draw Fig. 1. Source data for Figs. 2 and 3 are available as Supplementary Data 3 and 4.

## Code availability

All analyses were performed using the cited software, packages and pipelines, whose codes were publicly available.

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

## Acknowledgements

Istituto Buddista Italiano Soka Gakkai funded the project with the 8x1000 funds (ID n. 2020-2016_RIC_3). "PAT-COVID: Host genetics and pathogenetic mechanisms of COVID-19". The funding organization had no role in the design and conduct of the study; collection, management, analysis, and interpretation of the data; preparation, review, or approval of the manuscript. This study was a sub-project of the GEN-COVID consortium (https://sites.google.com/dbm.unisi.it/gen-covid; full list of GEN-COVID contributors is available as Supplementary Note 1). Specimens collected within the GEN-COVID consortium were provided by the COVID-19 Biobank of Siena, which is part of the Genetic Biobank of Siena, member of BBMRI-IT, Telethon Network of Genetic Biobanks (project no. GTB18001), Euro-BioBank, and RD-Connect.

## Author contributions

Conceptualization, F.C., M.Carella, M.F., and T.A.D.; formal analysis, M.E., F.M., M. Copetti. and F.C.; resources, A.R., S.C., M.Bruttini, M.Baldassarri, C.F., M.Carella, G.M., R.P., A.P., G.D.V., M.B., P.D.A, R.B., F.B, E.M.G.C, E.C, F.A., and R.E.M.; data curation, M.Baldassarri, M. Copetti, R.P., R.B., F.M, M.E., and F.C; writing—original draft preparation, F.C. and T.A.D.; writing—review and editing, T.A.D., F.C., M.F., C.F., P.C., and M. Copetti; funding acquisition, F.C., M.F., A.R., and M.Carella. All authors have read and agreed to the published version of the manuscript.

## Competing interests

The authors declare no competing interests.

## Additional information

¹National Research Council, Institute for Biomedical Technologies, Segrate, MI, Italy. ²Department of Medical Biotechnology and Translational Medicine (BioMeTra), Università degli Studi di Milano, Milan, Italy. ³Fondazione IRCCS Casa Sollievo della Sofferenza, San Giovanni Rotondo, FG, Italy. ⁴Med Biotech Hub and Competence Center, Department of Medical Biotechnologies, University of Siena, Siena, Italy. ⁵Medical Genetics, University of Siena, Siena, Italy. ⁶Genetica Medica, Azienda Ospedaliero-Universitaria Senese, Siena, Italy. ⁷Fondazione IRCCS Istituto Neurologico Carlo Besta, Milan, Italy. ⁸Aspidia srl, Milan, Italy. ⁹Istituto di Ricerche Farmacologiche "Mario Negri" IRCCS, Milan, Italy. ✉e-mail: francesca.colombo@cnr.it

