## [Peer Review File · Communications Medicine]

Reviewers' comments:

Reviewer #1 (Remarks to the Author):

Ref Submission ID COMMSMED-23-0483-T

In their paper entitled Multiple genetic control of anti-COVID-19 vaccine response by HLA locus, Esposito et al. performed a genome-wide association study in 1,351 individuals recruited from three Italian Hospitals to determine whether the antibody response to the SARS-CoV-2 Pfizer-BioNTech vaccine is genetically determined and what are the genetic factors involved in that response. They found several association signals in the HLA region, the strongest HLA-A. After imputation, HLA-A*03:01 was the most significantly associated with anti-SARS-CoV-2 spike antibody serum levels.

MAJOR COMMENTS

The manuscript is too descriptive, even in the Discussion. It reduced to commenting on the statistical results without any biological support. Specifically, the authors do not comment on their most surprising finding: a class I molecule that strongly influences an antibody-mediated response. Methodologically, the main question is that the manuscript does not include data about the ratio of infected (asymptomatic and symptomatic) individuals, which can generate bias and a misinterpretation of the results. This is an important confounder factor because infection is related to antibody levels and HLA-class I molecules with anti-viral response through their function in the presentation to the CD8 cells and controller of NK activity.

SPECIFIC COMMENTS: RESULTS and MATERIALS and METHODS

Table 1 and Results related

Please include data on infected (asymptomatic and symptomatic) individuals and the time from the infection to sampling. Incorporate these data into the regression models.

The time between vaccination and serological tests has differences among the three hospitals. It is 30 days in Milan and the San Giovanni Rotondo, but 97 days in the Siena Hospital. The authors should explain the reason for this difference in vaccination guidelines. Levels of antibodies depend on the time of vaccination, so how the different sampling schedules could affect the results? This question is relevant since the Siena cohort has more weight in the results because it includes more individuals than the other two.

Also, include the statistical results on demographic data comparison between groups as a footnote and discuss, if necessary, the bias they could introduce in the results.

Other parts of the Results and Materials and Methods sections related

Please cite in the Materials and Methods section "statistical analyses" the statistical tests used in the correlations (Results lanes 64-70). Also, describe the range of beta and define "SE".

GWAS data come from only one centre, but serological tests were performed in three centres using two different methods. How do they control differences among operators and tests?

Lane 66

The mean BAU/ml value of individuals from Siena is missing.

Please explain what the normalization of serological results consists of. Specifically, whether you transformed a quantitative into a categorical variable in the Material and Methods section and also as a footnote in the supplementary Figure 3

Lanes 71-74

Introduce how you construct the different regression models in the corresponding section of Materials and Methods. Specifically, Table 1 and data in lanes 64-70 suggest you have three dependent variables: the time of the second dose, the centre and the levels of antibodies. How do you manage them in the regression models?

Lane 75

Please change the sentence “A statistically significant locus...” to statistically significant signals...” or similar.

Lanes 79-91

The terms generate confusion. There are three independent signals (SNPs associated) but not three independent loci.

The SNP rs2499 has a much higher MAF than the other three associated SNPs. So, there are differences in statistical power. Please introduce the limitations of the study in the Discussion.

Lanes 92-100

What does it correspond to n=204? Is it the number of individuals with HLA four-digit data included in the reference panel? Please clarify.

You erroneously refer to “HLA-allele groups” as HLA-haplotypes. Also, you could use the expression “HLA molecules”.

You list several four-digit HLA molecules associated with the antibody levels with linkage disequilibrium in Europeans (C*12:02 with B*52:01 and DRB1*15:02 with DQB1*06:01). Comment in Discussion.

A limitation of this part is the different statistical power between molecules more and less common in the population.

What does it correspond to n=107? Is it the number of individuals with HLA two-digit data included in the reference panel? Please clarify. Could you explain why this size is lower than that for four digits?

The study at two and four digits is confusing because the low-resolution groups include the high

ones. After all, the difference is the level of resolution of the genotyping. This analysis could be misunderstood by non-expert HLA readers. Among other problems, some low-resolution groups are more homogenous than others. Thus, in Europeans, almost all the A*03 individuals will be A*03:01, but the individuals DQB1*06 will be DQB1*06:01, 06:02, 06:03, 06:04, 06:09. misunderstood. What contributes to the study of the separate analysis of low and high resolution? Please introduce all the limitations of the study.

Discussion

Rewrite it.

The authors must explain the mechanisms throughout the think a Class-I molecule could influence levels of antibodies in response to vaccination.

The study they cite (Bolze et al.) has a different approach, investigating adverse reactions to the vaccine. The mechanisms involved in toxicity and antibody production are dissimilar. Therefore, the results of the other team is not a support for theirs.

List and comment on the limitations.

Reviewer #2 (Remarks to the Author):

The paper “Multiple genetic control of anti-COVID-19 vaccine response by HLA locus”, by Esposito and colleagues, was designed to investigate the genetic control of response to COVID-19 immunization with Pfizer-BioNTech vaccine. The manuscript is well written and the study was designed and conducted using the appropriate methods to address this subject. The study provides new evidence for a key role of HLA locus on immune response to COVID-19 vaccination among Europeans.

Major comments:

- 1) The authors describe that antibody levels were lower among Siena participants (lines 66-69), but those results were expected since individuals were evaluated about 3 months after vaccination. I suggest the authors to perform the analysis independently for Siena and Milan+SGR cohorts to assess whether the results will be homogenous despite this difference in antibody levels;
- 2) The authors have showed results of a GWAS analysis and classic HLA alleles imputation. Have you considered performing SNP genotype imputation to increase genome-wide coverage?

Minor comments:

- 3) Please include r^2 values in lines 86-88.
- 4) Line 100: please verify the number of supplementary table provided in the Excel file.
- 5) On methods section, please explain the choice of the multi-ethnic panel for HLA imputation after removing the individuals who did not cluster with EUR samples from 1000G. What was the accuracy of your HLA imputation using this reference sample?
- 6) Figure 2 legend: please correct “... all the top...” on line 257.
- 7) Table 1: Please define SGR and BAU in table footnotes and use either range or IQR for all quantitative characteristics.

We wish to thank the Reviewers for the useful and thoughtful comments, that allowed us to better discuss our results.

Responses to Reviewers' comments

Reviewer #1

Question #1: Please include data on infected (asymptomatic and symptomatic) individuals and the time from the infection to sampling. Incorporate these data into the regression models.

Answer #1: Unfortunately, we do not have available data about infection of our individuals. However, we excluded from the study those subjects reporting a previous symptomatic SARS-CoV-2 (in the revised manuscript we specified this point at page 5, lines 93-94 and page 9, line 177). Additionally, vaccination guidelines provided for two administrations of anti-COVID-19 vaccine only to those who had not previous COVID-19 (and, thus, in absence of natural immunization), and all the individuals recruited in our study got the two doses. Therefore, we are sure to not have included previously symptomatic COVID-19 patients in our study. Additionally, most individuals recruited for this study were hospital workers and they were screened for SARS-CoV-2 infection every 15 days (before vaccination). Therefore, it is quite unlikely that any pre-infected asymptomatic individuals were enrolled in the study. For these reasons, we are confident that the measured antibody levels were induced by the vaccine and not by the natural infection.

Question #2: ...The authors should explain the reason for this difference in vaccination guidelines. Levels of antibodies depend on the time of vaccination, so how the different sampling schedules could affect the results?

Answer #2: The vaccination guidelines have been strictly followed in the three hospitals (the vaccine boost was administered three weeks after the first dose, to everyone). We clarified this point at page 4, lines 74-75 and page 9 line 178. The main difference between Siena and the other groups was in the interval time between the second dose and the measurement of antibody levels in serum and, for this reason, the results we reported were from a linear regression models in which this variable was included as covariate (page 7, lines 146 and page 8, lines 155-156, 162-168, 171-172).

Question #3: Also, include the statistical results on demographic data comparison between groups as a footnote and discuss, if necessary, the bias they could introduce in the results.

Answer #3: We added a column in the revised Table 1, reporting the statistical results requested. Anyway, the GWAS included the demographic variables as covariates, so the reported results were already corrected for these possible confounders.

Question #4: Please cite in the Materials and Methods section "statistical analyses" the statistical tests used in the correlations (Results lanes 64-70). Also, describe the range of beta and define "SE".

Answer #4: In the Methods section, we defined better the formula of the linear regression model used for this analysis (page 7, line 146). We moved the results of the analysis from the text to a new table (Table 2) where we defined SE as the standard error and added the 95% confidential interval.

Question #5: GWAS data come from only one center, but serological tests were performed in three centers using two different methods. How do they control differences among operators and tests?

Answer #5: We used the BAU/ml (Binding Antibody Unit/ml) values that was suggested for comparison between different serological tests (Infantino M, et al., Int Immunopharmacol. 2021). In addition, we reported the results of a GWAS that included the variable “center” as covariate in the regression model, to account for any possible confounding factors related to the operator/tests in the three recruiting centers. In the revised manuscript, we also added the results of the comparison of the IgG values measured at the same time point after vaccination (30 ± 5 days) among the three series, as we are aware that most of variability in IgG levels might be due to the time that passed between vaccination and IgG measurement. Kruskal-Wallis test indicated that there were no statistically significant differences between the patient series (page 10, lines 206-212), thus excluding possible confounders due to different operators and tests.

Question #6: (Lane 66) The mean BAU/ml value of individuals from Siena is missing.

Answer #6: In Table 1, we reported median values of antibodies (in BAU/ml units) in the whole series and in the three cohorts.

Question #7: Please explain what the normalization of serological results consists of. Specifically, whether you transformed a quantitative into a categorical variable in the Material and Methods section and also as a footnote in the supplementary Figure 3.

Answer #7: We did not transform the quantitative antibody variable into a categorical one. In the revised version of the manuscript, we explained better the use of the inverse normal transformation, at page 7, lines 139-142.

Question #8: (Lanes 71-74) Introduce how you construct the different regression models in the corresponding section of Materials and Methods. Specifically, Table 1 and data in lanes 64-70 suggest you have three dependent variables: the time of the second dose, the centre and the levels of antibodies. How do you manage them in the regression models?

Answer #8: We performed a multivariate linear regression model to identify the variables affecting the levels of antibodies. This was the model formula:

$$\text{Normalized Ab} \sim \text{age} + \text{sex} + \text{time between second dose and serological test} + \text{center}$$

The formula of the GWAS regression model was:

$$\text{Normalized Ab} \sim \text{genotype} + \text{age} + \text{sex} + \text{time between second dose and serological test} + \text{center} + \text{PC1} + \text{PC2} + \text{PC3} + \text{PC4} + \text{PC5}$$

We added these formulas to the revised Methods section (page 7, lines 146 and page 8, lines 155-156).

The variables *age*, *time between second dose and serological test*, and the first five *principal components* (PCs) were included in the model as numeric ones, whereas *sex* was a binary variable and *center* was coded as a dummy variable. The three genotype classes were considered as in an additive model (i.e., coded as 0, 1, 2 according to the increasing number of minor alleles).

Please, note, that as explained above, we did not include in the model the time of the second dose, that was equal for every subject, but the time between the second dose and the serological test (pages 7-8, lines 151-154).

Question #9: (Lane 75) Please change the sentence “A statistically significant locus...” to statistically significant signals...” or similar. ... (Lanes 79-91) The terms generate confusion. There are three independent signals (SNPs associated) but not three independent loci.

Answer #9: We edited the terms as suggested.

Question #10: The SNP rs2499 has a much higher MAF than the other three associated SNPs. So, there are differences in statistical power. Please introduce the limitations of the study in the Discussion.

Answer #10: In our series rs2499 had a MAF=11% while the other three variants had MAF=10%, 14% and 8.7%, respectively. We do not think that these small differences could severely affect the statistical power and the results of our study. Please note that, in response to Reviewer #2, we carried out a new GWAS on imputed genotypes (and the same linear regression model). Therefore, in the revised manuscript we updated our results, with the top-significant SNP now being rs1632893. We reported the MAF of the top SNPs (rs2499 is among the top 15) in the revised Supplementary Table 1.

Question #11: (Lanes 92-100) What does it correspond to n=204? Is it the number of individuals with HLA four-digit data included in the reference panel? Please clarify.

Answer #11: We clarified that the number 204 referred to the imputed four-digit HLA-allele groups (page 8, lines 169 and page 12, line 256) and not to the subjects included in the analyses (n = 1,351).

Question #12: You erroneously refer to “HLA-allele groups” as HLA-haplotypes. Also, you could use the expression “HLA molecules”.

Answer #12: We edited the terms as suggested.

Question #13: You list several four-digit HLA molecules associated with the antibody levels with linkage disequilibrium in Europeans (C*12:02 with B*52:01 and DRB1*15:02 with DQB1*06:01). Comment in Discussion.

Answer #13: Our LD data agreed with what indicated by the reviewer. We added these results (page 12, lines 268-269 and Supplementary Table 3) and briefly discussed them (page 14, lines 304-306).

Question #14: A limitation of this part is the different statistical power between molecules more and less common in the population.

Answer #14: We are aware that the study had no sufficient statistical power to identify significant association with rare HLA alleles. However, we were able to detect significant associations (FDR<0.01) with alleles with a frequency higher than 2% (Supplementary Table 3, page 12, lines 265-268 and page 14, lines 303-304).

Question #15: What does it correspond to n=107? Is it the number of individuals with HLA two-digit data included in the reference panel? Please clarify. Could you explain why this size is lower than that for four digits?

Answer #15: Just for sake of clarity, the number 107 referred to the imputed two-digit HLA-allele groups. However, to avoid confusion, we decided to report only the results of four-digit HLA-allele groups (page 12, lines 256-269).

Question #16: The study at two and four digits is confusing because the low-resolution groups include the high ones. After all, the difference is the level of resolution of the genotyping. This analysis could be misunderstood by non-expert HLA readers. Among other problems,

some low-resolution groups are more homogenous than others. Thus, in Europeans, almost all the A*03 individuals will be A*03:01, but the individuals DQB1*06 will be DQB1*06:01, 06:02, 06:03, 06:04, 06:09. misunderstood. What contributes to the study of the separate analysis of low and high resolution? Please introduce all the limitations of the study.

Answer #16: To avoid confusion, we decided to report only the results of four-digit HLA-allele groups (page 12, lines 256-269).

Question #17: The authors must explain the mechanisms throughout the think a Class-I molecule could influence levels of antibodies in response to vaccination.

Answer #17: We are aware that HLA class I alleles are not directly involved in humoral response. Indeed, we think that HLA-DQB1, HLADQA1, or HLA-DRB1 alleles, rather than HLA-A*03:01 and HLA-C*12:02, may be a susceptibility genetic factor directly affecting antibody-mediated response to anti-COVID-19 vaccination and IgG levels. However, based on our results, we hypothesize that class-I molecules may be an additional genetic factor underlying the individual immunological response to the vaccine. Indeed, this molecule can influence the susceptibility to induce a more or less potent CD8+ T cell response following vaccination, in turn affecting the overall individual response to the vaccine. According with this idea, SARS-CoV-2 mRNA vaccination has been reported to induce both neutralizing antibodies and CD8+ T cell responses, with the last having relevant protective effects as well as the ability to affect B cell response (Oberhardt, V. et al. Nature 2021; Chen et al. Biology 2023; Tyllis et al. Front Immunol 2021). Moreover, both CD8+ T and B cell responses have been associated with protection against SARS-CoV-2 following mRNA vaccination (Brasu et al. Nature Immunology 2022), thus supporting a contribution of HLA-A*03:01, along with HLA-DQB1, to the vaccine-mediated protection. Future studies are needed to deepen the relationship between HLA-A*03:01 and CD8+ T cell response upon anti-COVID-19 vaccination. We added a comment on this issue in the Discussion of the revised manuscript.

Question #18: The study they cite (Bolze et al.) has a different approach, investigating adverse reactions to the vaccine. The mechanisms involved in toxicity and antibody production are dissimilar. Therefore, the results of the other team is not a support for theirs.

Answer #18: We revised the sentence (page 14, lines 297-301).

Question #19: List and comment on the limitations.

Answer #19: As requested, we widely rewrite the Discussion section, commenting on the limitations of the study.

Reviewer #2

Question #1: I suggest the authors to perform the analysis independently for Siena and Milan+SGR cohorts to assess whether the results will be homogenous despite this difference in antibody levels

Answer #1: We performed the additional analysis suggested by the Reviewer and here below we show the Manhattan plots of the GWAS performed in Siena (A) and Milan+SGR (B) cohorts.

As expected, both analyses had a lower power than the analysis reported in the manuscript. The signal at chromosome 6 (HLA locus) was evident in both analyses, though with lower P -values, particularly in the Milan+SGR cohort. This might be explained by the higher genetic heterogeneity of this cohort (including individuals from Northern and Southern Italy) than the Siena cohort.

Zooming in HLA locus, we observed that the leading variant in Siena cohort (A) was in *HLA-B* locus; other variants above the genome-wide significance threshold mapped in *HLA-DQB1* locus. Variant near *HLA-A* gene were just below the threshold. In Milan+SGR cohort (B) the lead variant (P -value = 8.88×10^{-7}) mapped ~40kb ahead *HLA-A* gene.

Therefore, we decided to show in the manuscript the results from the comprehensive GWAS analysis (including all the 1,351 subjects, and corrected for “center” variable) that, in our

opinion, is the most informative since it showed all the three main signals observed in the separated analyses.

In addition to the Reviewer's suggestion, we also carried out another GWAS on 574 subjects whose IgG levels were homogeneously measured 30 ± 5 days after the second dose of vaccination. In this subset, no significant differences in IgG levels were observed among the three centers (page 10, lines 206-212). Here below we reported the results of this further analysis:

The limited number of samples in the analysis reduced the statistical power and the signal on chromosome 6 was less evident. Nonetheless, the most significant signal was at *HLA-A* locus). We decided not to show these results since we think that this analysis is underpowered.

Anyway, the linear regression model of the GWAS, whose results were reported in the manuscript, included the time interval between second vaccination dose and serological test, as covariate, to control for this possible confounder (see also Methods section, page 8, lines 155-156).

Question #2: Have you considered performing SNP genotype imputation to increase genome-wide coverage?

Answer #2: In the revised manuscript, we reported the results from a new GWAS with an imputed dataset that included genotype data from 7,339,393 markers. Methods and Results sections, as well as Figures, were updated accordingly.

Question #3: Please include r^2 values in lines 86-88.

Answer #3: We added r^2 as requested (page 11, lines 243-245). LD measures were also reported in Supplementary Table 1.

Question #4: Line 100: please verify the number of supplementary table provided in the Excel file.

Answer #4: We updated the Supplementary Table numbers.

Question #5: On methods section, please explain the choice of the multi-ethnic panel for HLA imputation after removing the individuals who did not cluster with EUR samples from 1000G. What was the accuracy of your HLA imputation using this reference sample?

Answer #5: On the Michigan Imputation server, only multi-ethnic reference panels are available for HLA imputation. Variants with an imputation score (R^2) < 0.7 were filtered out. We added this

information about the imputation accuracy filter in the Methods section of the revised manuscript (page 7, lines 128).

Question #6: Figure 2 legend: please correct "... all the top..." on line 257.

Answer #6: We edited the text, as indicated.

Question #7: Table 1: Please define SGR and BAU in table footnotes and use either range or IQR for all quantitative characteristics.

Answer #7: In the revised Table 1 we added the footnotes with definitions and used IQR for the quantitative variables (except for age which is usually reported as median and range).

REVIEWERS' COMMENTS:

Reviewer #1 (Remarks to the Author):

The authors have answered all my questions and have done an adequate and exhaustive review of the original manuscript. I have no further comments to make. In my opinion, the revised version is suitable for publication.

Reviewer #2 (Remarks to the Author):

Recommendation: accept.

All comments and suggestions were addressed. The author's decision not to incorporate some of the new analyses I have suggested was explained in details.